# A Fraction of Recommended Practices: Implementation of the FIFA 11+ in NCAA Soccer Programs

**DOI:** 10.3390/medicina56090417

**Published:** 2020-08-19

**Authors:** Lawrence W. Judge, Jeffrey C. Petersen, Donald L. Hoover, Bruce W. Craig, Nick Nordmann, Makenzie A. Schoeff, Brian D. Fox, D. Clark Dickin, David M. Bellar

**Affiliations:** 1School of Kinesiology, Ball State University, Muncie, IN 47306, USA; bcraig@bsu.edu (B.W.C.); nicholasnordmann2015@gmail.com (N.N.); maschoeff@bsu.edu (M.A.S.); bdfox@bsu.edu (B.D.F.); dcdickin@bsu.edu (D.C.D.); 2School of Education, Baylor University, Waco, TX 76798, USA; Jeffrey_Petersen@baylor.edu; 3Department of Physical Therapy, Western Michigan University, Kalamazoo, MI 49008, USA; don.hoover@wmich.edu; 4School of Kinesiology, University of North Carolina at Charlotte, Charlotte, NC 28223, USA; dbellar@uncc.edu

**Keywords:** athletic performance, cool-down, non-contact injury, training theory, warm-up

## Abstract

*Background and Objectives:* National Collegiate Athletic Association (NCAA) soccer coaches implement numerous warm-up and flexibility strategies to prepare athletes for training and competition. The Fédération Internationale de Football Association (FIFA) developed the 11+ injury prevention program to reduce non-contact injuries. This study aimed to analyze the level of familiarity with and implementation of the evidence-based FIFA 11+ amongst NCAA Division I (DI) and Division III (DIII) men’s and women’s soccer coaches. *Materials and Methods:* NCAA soccer coaches in the United States received an Institutional Review Board—approved survey hyperlink. A total of 240 coaches completed the survey. The respondents represented 47.5% men’s and 52.5% women’s teams distributed within DI and DIII programs. Descriptive statistics are reported as frequency counts and mean ± standard deviation where applicable. Pearson’s chi-square tests were performed to assess potential differences with a significance level set at α < 0.05. *Results:* The results indicated that approximately 62% of the respondents reported being familiar with the FIFA 11+ program. Of those coaches familiar with the program, 15.0% reported full implementation, 57.5% reported partial implementation, and 27.5% reported no implementation. Chi-square analyses revealed significant differences in FIFA 11+ implementation based upon division level (χ^2^ = 4.56, *p* = 0.033) and coaching certification levels (χ^2^ = 13.11, *p* = 0.011). *Conclusions:* This study indicates that there is a gap between FIFA 11+ knowledge and actual implementation. To reduce the risk of non-contact injury, there is a need to educate coaches and athletic trainers on the purpose of the FIFA 11+ program and how to perform the exercises correctly.

## 1. Introduction

Soccer is the most popular sport played worldwide, with recent estimates of approximately 200,000 professional and 240 million amateur players [1]. In the United States, soccer participation has continued to increase in popularity [2], especially in the National Collegiate Athletic Association (NCAA), with more than 50,000 American collegiate players [3]. The increase in the number of participants has led to multiple emotional and physical health and wellness benefits [4]. However, there are risks associated with soccer participation, and soccer-related injuries are not uncommon [4]. More than 8000 injuries were reported in a cohort of NCAA Division I, Division II, and Division III male and female soccer programs from 2004 to 2009 via the injury surveillance system [5]. There is a growing body of evidence that suggests reductions in the incidence of soccer injuries in both men and women can be observed by implementing injury prevention programs [4,6,7,8].

The Fédération Internationale de Football Association (FIFA) developed the FIFA 11+ injury prevention program in cooperation with the FIFA Medical Assessment and Research Centre (F-MARC), the Santa Monica Orthopedic and Sports Medicine Research Foundation (SMSMF), and the Oslo Sports Trauma Research Centre (OSTRC), to prevent non-contact injuries and promote soccer as a health-enhancing physical activity [9,10]. The FIFA 11+ injury prevention program is comprised of 15 total exercises with a focus on core stabilization, neuromuscular control and balance, eccentric training of the hamstrings, plyometrics, and agility [11]. The program includes preventative interventions such as the improvement of the warm-up and cool-down [10] and can be completed in 10–15 min on the field without any additional technical equipment [6]. 

Various studies have reported using the FIFA 11+ program with a reduced risk of injury in soccer players ranging from 41% to 72% [4,7,12,13]. However, two additional studies that analyzed the efficacy of the 11+ warm-up failed to indicate a significant reduction in injury rate, which was likely a factor in the low overall compliance with the program [14,15]. There must an adequate overall number and frequency of neuromuscular training sessions to derive the preventative effects of the warm-up [14]. A study examining the efficacy of the FIFA 11+ injury prevention program among NCAA Division I and II male soccer players found a statistically significant reduction in injuries in the intervention group and an inverse relationship between injury rate and compliance [4]. As compliance with the program increased, the number of sport-specific-related injuries decreased with statistical significance (*p* = 0.034). These data reinforce the findings of other authors and highlight the importance of the regularly scheduled implementation of and adherence to injury prevention programs. An additional study by Grooms et al. [7] evaluated the effects of the FIFA 11+ injury prevention program on lower extremity injury incidence in male collegiate soccer players. This study’s results indicated that the 11+ program may be more effective at decreasing the incidence of lower extremity injury and time lost from sport than traditional pre-activity warm-up practices. In addition to reducing the risk of injury, the 11+ program has been validated as an effective warm-up for inducing positive acute physiological responses that can enhance sprinting performance, agility, and vertical jump ability [16]. 

The effectiveness of the FIFA 11+ warm-up for enhancing performance depends on the team’s compliance, which is a factor in the motivation, decisions, and actions of the head coach [13]. The head soccer coach is often responsible for implementing pre-activity warm-up and stretching practices at the collegiate level. However, some of these coaches may be unaware of the current recommendations regarding injury prevention programs [17]. An examination of the FIFA 11+ injury prevention program’s level of implementation among Australian and Saudi Arabian professional and semi-professional soccer coaches revealed a substantial gap between the FIFA 11+ program recommendations and the actual practices among soccer coaches [18]. In Australia, 73% of the coaches implemented the FIFA 11+ program. However, only 51% of those coaches implemented the entire FIFA 11+ exercise components as recommended. In Saudi Arabia, only 40% of the coaches followed the FIFA 11+ program [18]. Thus, there is a need for the comprehensive education of coaches with regard to the importance of appropriate implementation of the FIFA 11+ warm-up. According to Steffen et al. [19], proper coaching education plays a substantial role in team adherence to injury prevention programs. Their study results indicate that comprehensive coach education at the beginning of the season, including a practical workshop, was more effective at improving adherence to the 11+ than a web-based delivery of the program. The reported injury rates support the notion that the risk of overall injury is reduced with high adherence to 11+ implementation. Therefore, coach education should play an important role in the way the FIFA 11+ warm up is delivered to teams and individual players.

A focus on the implementation of injury prevention programs in football and professional development for coaches has continued to grow in settings outside of American collegiate soccer. This has included studies of high school programs [20], professional youth teams [21,22], female adolescents [23,24], club-level teams [25], and a nation-wide effort across amateur teams [26]. These prior studies tended to focus upon developmental (youth or amateur) programs, although a dichotomy of professional versus developmental was noted within the professional youth setting. NCAA soccer could also be interpreted along such a dichotomy, with DI programs providing multi-year, cost-of-attendance athletic scholarships being viewed as more professionalized compared with the DII programs providing no athletic scholarship support. 

Although previous research has identified a gap between soccer coaches’ knowledge with regard to injury prevention programs and their actual practice [17], no research has been conducted to assess the implementation strategies for the FIFA 11+ exercise components and the level of familiarity among NCAA Division I and Division III soccer coaches in the United States. The findings of such investigations could influence the adherence among teams and individual players. Therefore, the purpose of the present study was to investigate the knowledge and practices of NCAA Division I and Division III men’s and women’s soccer coaches to determine the level of familiarity and implementation of the evidenced-based FIFA 11+ warm-up. Based on previous research, it was hypothesized that there would be a high level of familiarity with the FIFA 11+ program and moderate level of implementation. It was further hypothesized that there would be a significant difference in the level of implementation based upon NCAA division classification and coaching certification level.

## 2. Materials and Methods

This investigation used a descriptive survey study design. An online survey instrument allowed data collection from NCAA Division I and Division III men’s and women’s soccer coaches. The survey provided insight into the knowledge and pre-activity stretching and warm up practices implemented by collegiate coaches in the United States. The data were used to understand the level of familiarity and implementation of the FIFA 11+ injury prevention program.

### 2.1. Subjects 

NCAA Division I or Division III coaches participated in this study. Eligible participants met the following inclusion criteria: (1) head or assistant coach of a men’s or women’s soccer program in the United States and (2) at least 18 years of age. Before completing the survey, the subjects were informed of the purpose, risks, and benefits of the investigation. Participants meeting all the inclusion criteria gave their informed consent in accordance with the Declaration of Helsinki. A total of 239 NCAA coaches representing (47.5%) men’s and (52.5%) women’s teams distributed within (22.9%) DI and (77.1%) DIII programs completed the survey, and this represents approximately 17.5% of all the teams within these groups. The mean age of the 239 respondents was 44.8 ± 10.4 years, and the mean coaching experience was 21.9 ± 9.1 years (see Table 1). The appropriate Institutional Review Board approved the survey instrument and research protocols.

### 2.2. Procedures 

A hyperlink to the online survey was sent to all NCAA Division I and Division III men’s and women’s soccer coaches in the United States. The brief online research questionnaire was adapted from survey instruments used previously in similar studies [27,28,29]. The final modified survey instrument consisted of 17 multiple-choice or open-ended questions covering demographic information (6 items), coach demographic information (3 items), pre-activity warm-up information (2 items), and questions pertaining to the FIFA 11+ (6 items). To ensure clarity and content, the survey questionnaire was reviewed by a licensed U.S. Soccer Federation coach. An e-mail invitation was sent to all non-respondents in an attempt to increase the response rate.

### 2.3. Statistical Analyses 

The validity of the survey questionnaire was analyzed using principal component analysis. Using Kaiser–Meyer–Olkin statistics, like items on the survey were compared and sampled for similarity. The results indicate that the instrument had a construct validity >0.70. The data were collectively gathered, and descriptive statistics were used to determine the frequencies and measures of central tendency when applicable. Statistical analyses were performed using JMP version 13.0, and the criterion for significance for all analyses was set at α < 0.05. A power analysis was conducted a priori using previous literature published by the co-authors of this study [27,28,29]. The calculation suggested that the current sample size was adequate.

## 3. Results

### 3.1. Current Pre-Activity Practices 

The overwhelming majority of the coaches in this survey (99.1%) reported using some form of general warm-up, consisting of jogging (12.3%), mini games (28.6%), soccer-specific drills (28.6%), or other related activities (29.5%). Almost all the coaches (99.6%) reported the use of pre-activity stretching practices, while only one coach reported not having their athletes perform any pre-activity stretching. The majority of the coaches (85.9%) implemented dynamic stretching prior to activity. Nineteen of the coaches (8.4%) reported the use of proprioceptive neuromuscular facilitation pre-activity, with seven coaches (3.1%) indicating “other” as part of their pre-activity practices. The least popular pre-activity stretching practice prior to an athletic event was static stretching (0.0%), followed by ballistic stretching (2.2%).

### 3.2. FIFA 11+ Familiarity and Implementation

The results indicated that 61.9% of the respondents reported being familiar with the FIFA 11+ program. Of those coaches familiar with the FIFA 11+ program, 15.0% reported full implementation, 57.5% reported partial implementation, and 27.5% reported no implementation. Figure 1 presents the frequency count of implementation for each part of the FIFA 11+ program.

The level of coaching (Division I vs. Division III) was determined to significantly influence FIFA 11+ implementation (χ^2^ = 6.27, *p* = 0.012). The majority of Division I coaches (77.3%) were familiar with the FIFA 11+ program, with 13.6% of those coaches reporting full implementation and 45.5% reporting partial implementation. Of the Division III coaches, 57.0% were familiar with the FIFA 11+ program, with 8.2% of those coaches reporting full implementation and 33.5% reporting partial implementation. Chi-square analyses also revealed significant differences in FIFA 11+ implementation based upon coaching certification levels (χ^2^ = 24.75, *p* < 0.001). A significantly higher number of coaches (52.7%) with one to three soccer certifications reported not being familiar with the FIFA 11+ program in comparison to coaches with four to six certifications (34.1%), α < 0.05. Furthermore, coaches with one to three soccer certifications (26.0%) were significantly less likely to partially implement the FIFA 11+ program in comparison to coaches with four to six certifications (40.7%) α < 0.05 (Table 2).

Neither FIFA 11+ familiarity (χ^2^= 0.056, *p* = 0.814) nor implementation (χ^2^ = 0.248, *p* = 0.618) were significantly impacted by the gender of the team. The level of familiarity with the FIFA 11+ program for both the men’s (60.4%) and women’s (62.3%) soccer teams was relatively similar. Of the coaches for the men’s teams, seven (7.3%) reported full implementation, 35 (36.5%) reported partial implementation, and 16 (16.7%) reported no implementation. Of the coaches for the women’s teams, 12 (11.3%) reported full implementation, 38 (35.9%) reported partial implementation, and 19 (17.9%) reported no implementation. 

The gender of the coach was not determined to significantly influence FIFA 11+ familiarity or implementation (χ^2^ = 1.38, *p* = 0.24; χ^2^ = 1.18, *p* = 0.75, respectively). Of the male respondents, 60.5% were familiar with the FIFA 11+ program. Fifteen male coaches (9.0%) reported full implementation, 60 (35.9%) reported partial implementation, and 28 (16.8%) reported no implementation. Of the female respondents, 70.6% were familiar with the FIFA 11+ program. Four female coaches (11.8%) reported full implementation, 13 (38.2%) reported partial implementation, and seven (20.6%) reported no implementation. The results of the chi-square analysis did not determine age (χ^2^ = 6.44, *p* = 0.09) nor years of coaching experience (χ^2^ = 2.68, *p* = 0.44) to significantly impact the level of familiarity with or implementation of the FIFA 11+ program. Coaches reported no implementation of the FIFA 11+ for the following reasons: not familiar (42.6%), prefer to use their own warm up (18.8%), lack the time (6.3%), lack the knowledge of the FIFA 11+ to implement it (2.8%), or other (29.5%). 

## 4. Discussion

This study aimed to determine the familiarity with and implementation of the FIFA 11+ warm-up among NCAA Division I and Division III soccer coaches. There were two hypotheses for this study: First, there would be a high level of familiarity with and a moderate level of implementation of the FIFA 11+ program among coaches. Second, differences would exist between both division levels and coaching certification levels. The first hypothesis was supported; of the 230 coaches surveyed, over half of them (61.9%) reported being familiar with the FIFA 11+ program, and of those coaches, approximately two-thirds of them (66.1%) reported implementing the 11+ program in some way. The second hypothesis was also supported; over three-quarters of Division I coaches (77.3%) reported familiarity with the 11+ program, and over half of those coaches (59.1%) reported implementing it into their own programming in some fashion, whereas 57.0% of Division III coaches were familiar with the program, and less than half of them (41.7%) reported any sort of implementation. Similarly, coaches with one to three coaching certifications were more likely to report not being familiar with the 11+ program (52.7%) compared to coaches with four to six certifications (34.1%) and were also less likely to implement the program into their own training (26.0%) than coaches with four to six certifications (40.7%).

The FIFA 11+ program is designed to provide players with a dynamic, preactivity warm-up that serves to reduce injury without requiring additional on-field equipment [6]. It has also been demonstrated to provide players with physiological responses that acutely enhance sprinting ability, agility, and vertical jump ability [16]. Thus, the 11+ program has the potential to serve as not only an injury prevention program but also as an efficient on-field warm-up prior to competition. As the inclusion of any type of preactivity warm-up was commonplace for most coaches in this study, and lack of familiarity was the most frequent reason given for not implementing the program (42.6%), it may be beneficial for soccer coaches better to familiarize themselves with the 11+ program for possible inclusion.

In the present study, 99.1% of the coaches surveyed reported including some type of warm-up in their practices, with dynamic warm-ups being the most favored among these coaches (85.9%), which is consistent with other studies conducted by the authors [20,21]. Interestingly, as much debate exists regarding the inclusion of static stretching as a warm-up, no coaches in this study reported including static stretching as a part of their preactivity. Coaches also mentioned that jogging (12.3%) and sport-specific activities (28.6%) were common inclusions in warm-up routines, which could be important regarding the integration of the 11+ program; a majority of the activities present in the 11+ program are either jogging or sport-specific activities [11].

Injury prevention is another integral part of the 11+ program. Multiple studies have demonstrated that integration of (and subsequent adherence to) the 11+ program reduces both risk for injury [4,7,12,13] and injury incidence [4,7]. While some studies suggest that this program had no effect on injury risk [14,15], their null findings may be explained by a lack of adherence to the program. These findings are equally important to those demonstrating a positive effect, as they reinforce the necessity for the program to be implemented in its entirety. With half of the coaches reporting partial implementation of the program into their routines, it is possible that complete implementation could yield additional benefits.

One possible explanation for the lack of familiarity and inclusion among Division III coaches is a lack of accessibility and social connection with the methods [30]. Steffen et al. [19] mentions that in-person training for the coaches increased adherence to the 11+ program throughout the season compared to that achieved with the online sessions, and coaches who had increased adherence to the program saw decreases in injury throughout the season. Thus, improving accessibility to the in-person trainings would allow Division III coaches to better implement the 11+ program into their routines. Better access would also allow for coaches with fewer certifications to have greater exposure to the program, as they tended to be less familiar with the program compared to coaches with more certifications.

One potential limitation of the study was the distribution of Division I to Division III coaches that responded to the study. Division III coaches were represented approximately three times more than Division I coaches. It is possible that Division I coaches are more frequently solicited for surveys and must be more selective with the surveys they respond to. This did not affect the interpretation of the answers, but it may not have provided an adequate representation of the familiarity of Division I coaches with the 11+ warm-up. Another limitation is that Division II coaches were not surveyed in the present study due to the researchers’ desire to more clearly differentiate the professional/amateur dichotomy between the DI and DIII levels, and thus, the results of this study cannot be universally applied to all NCAA coaches/trainers. Future studies should attempt to gain an even representation of Division I, II, and III coaches/trainers to gain a better idea of familiarity and implementation between divisions. The assessment of the many types of coaching certification is another potential source of limitation. Finally, the study design was a descriptive analysis and cannot answer questions about the impact of the use of FIFA 11+ within these programs.

## 5. Conclusions

Based on the results of this study, almost two-thirds of the surveyed coaches reported being familiar with the FIFA 11+ program. Coaches and athletic trainers should acknowledge this and consider familiarizing themselves with the program and including similar movements into the warm-ups and/or exercises for the athletes to become comfortable with the movement patterns, thus allowing for more seamless integration.

## Figures and Tables

**Figure 1 medicina-56-00417-f001:**
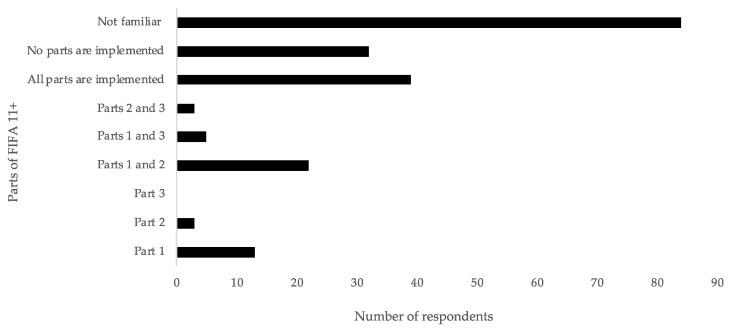
Count response of FIFA 11+ implementation.

**Table 1 medicina-56-00417-t001:** Participant characteristics.

Variable	Descriptor	Percentage
Sex	Male	80.0 (*n* = 191)
	Female	16.0 (*n* = 39)
	Prefer not to respond	4.0 (*n* = 10)
Soccer experience	Professional	31.7 (*n* = 76)
	Collegiate	59.6 (*n* = 143)
	High School	0.4 (*n* = 1)
	Club	2.9 (*n* = 7)
	None	1.2 (*n* = 3)
	Prefer not to respond	4.2 (*n* = 10)
Coaching experience	Head Coach	98.3 (*n* = 236)
	Assistant Coach	1.3 (*n* = 3)
	Prefer not to respond	0.4 (*n* = 1)
Number of soccer coaching certification(s)	0	0.8 (*n* = 2)
	1–3	34.5 (*n* = 83)
	4–6	44.6 (*n* = 107)
	7–9	9.2 (*n* = 22)
	10 or more	6.7 (*n* = 16)
	Prefer not to respond	4.2 (*n* = 10)

**Table 2 medicina-56-00417-t002:** Count response of FIFA 11+ implementation.

Number of Coaching Certifications	Descriptor	Group Percentage	Total Percentage
1–3	Not Familiar	52.1 (*n* = 38)	36.32 (*n* = 73)
	Full Implementation	6.9 (*n* = 5)	
	Partial Implementation	26.0 (*n* = 19)	
	No Implementation	15.1 (*n* = 11)	
4–6	Not Familiar	33.0 (*n* = 30)	45.27 (*n* = 91)
	Full Implementation	8.8 (*n* = 8)	
	Partial Implementation	40.7 (*n* = 37)	
	No Implementation	17.6 (*n* = 16)	
7–9	Not Familiar	4.8 (*n* = 1)	10.45 (*n* = 21)
	Full Implementation	9.5 (*n* = 2)	
	Partial Implementation	61.9 (*n* = 13)	
	No Implementation	23.8 (*n* = 5)	
10 or more	Not Familiar	21.4 (*n* = 3)	6.97 (*n* = 14)
	Full Implementation	28.6 (*n* = 4)	
	Partial Implementation	28.6 (*n* = 4)	
	No Implementation	21.4 (*n* = 3)	

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
