# Peer review of "A Fraction of Recommended Practices: Implementation of the FIFA 11+ in NCAA Soccer Programs"

_medicina, 2020, doi:10.3390/medicina56090417_

Round 1

Reviewer 1 Report

This study provides important information that will help advance care for athletes.  It is generally well written, although there are formatting and syntax errors throughout that need detailed proofreading and corrections (lines 4, 51, 65, 176 and others).

Study of DI and III and not II needs to be further justified in Introduction and Discussion.

While it is alluded to, the limitations surrounding selection bias of participants should be more fully described.

Limit the conclusions to what you have data for - the conclusions regarding training methods are not supported by data/evidence.

Author Response

Reviewer 1

This study provides important information that will help advance care for athletes.  It is generally well written, although there are formatting and syntax errors throughout that need detailed proofreading and corrections (lines 4, 51, 65, 176 and others).

Response: Read through the paper and made adjustments to formatting and syntax as requested.

Study of DI and III and not II needs to be further justified in Introduction and Discussion.

Response: The NCAA divisional membership includes three levels (I, II & III), and within those divisions the greatest number of schools and athletes lies within DIII (n = 438) 40% with 39% of the total student athletes. DI follows with 350 schools (32%) including 37% of the total athletes.  DII lags behind with the smallest number of schools and athletes.  Additionally, DII provides partial scholarship funding compared with the higher levels of funding provided with the DI level and the complete lack of athletic scholarships provided at the DIII level. The ability to differentiate between a collegiate system that is more professionalized (DI) to one that would be deemed far more amateur (DIII) was one rationale behind this sample selection. Inclusion of DII was considered to be a possible confounding condition between this professional/amateur dichotomy.

Additionally, the Silvers et al. (2014) study was conducted with male collegiate programs in DI & DII but did not assess DIII. This study thereby covered a prior gap with the inclusion of DIII programs.

A portion of this rationale has been inserted in the introduction and within the limitation section of discussion portion of the manuscript.

While it is alluded to, the limitations surrounding selection bias of participants should be more fully described.

Response: Added further rationale as to the participants that were surveyed, as well as suggestions for future studies.

Limit the conclusions to what you have data for - the conclusions regarding training methods are not supported by data/evidence.

Response: Removed training method conclusions.

Reviewer 2 Report

Thank you for the opportunity to review this manuscript. while the introduction provides an adequate summary of the literature on the effectiveness of the 11+, neither the introduction or discussion situate this study in the relevant literature related to the implementation of the 11+ or other injury prevention programs in football.

Norcross MF, Johnson ST, Bovbjerg VE, Koester MC, Hoffman MA. Factors influencing high school coaches’ adoption of injury prevention programs. J Sci Med Sport. 2016;19(4):299–304.

O'Brien J, Finch CF. Injury prevention exercise programmes in professional youth football: understanding the perceptions of programme deliverers. BMJ Open Sport Exerc Med. 2016;2(1):e000075.

Morgan EA, Johnson ST, Bovbjerg VE, Norcross MF. Associations between player age and club soccer coaches’ perceptions of injury risk and lower extremity injury prevention program use. Int J Sports Sci Coach. 2018;13(1):122–8.

Junge A, Lamprecht M, Stamm H, Hasler H, Bizzini M, Tschopp M, Reuter H, Wyss H, Chilvers C, Dvorak J. Countrywide campaign to prevent football injuries in Swiss amateur players. Am J Sports Med. 2011;39(1):57–63.

Lindblom H, Waldén M, Carlfjord S, Hägglund M. Implementation of a neuromuscular training programme in female adolescent football: 3-year follow-up study after a randomised controlled trial. Br J Sports Med. 2014;48(19):1425–1430.

Joy EA, Taylor JR, Novak MA, Chen M, Fink BP, Porucznik CA. Factors influencing the implementation of anterior cruciate ligament injury prevention strategies by girls football coaches. J Strength Condit Res. 2013;27(8):2263–2269.

Stoszkowski J, Collins D. Sources, topics and use of knowledge by coaches. J Sports Sci. 2016;34(9):794–802.

O'Brien J, Young W, Finch CF. The use and modification of injury prevention exercises by professional youth football teams. Scand J Med Science Sports. 2016. doi: 10.1111/sms.12756

Author Response

Reviewer 2

Thank you for the opportunity to review this manuscript. while the introduction provides an adequate summary of the literature on the effectiveness of the 11+, neither the introduction or discussion situate this study in the relevant literature related to the implementation of the 11+ or other injury prevention programs in football.

Response:

Indeed, the broad perspective of injury prevention programs within football were not fully addressed within the original submission and the recommendation of several key studies have now been integrated within the latter portion of the introduction and within the discussion section of the paper.  Thank you for the suggested studies for inclusion, all have been integrated into the manuscript.

Norcross MF, Johnson ST, Bovbjerg VE, Koester MC, Hoffman MA. Factors influencing high school coaches’ adoption of injury prevention programs. J Sci Med Sport. 2016;19(4):299–304.

O'Brien J, Finch CF. Injury prevention exercise programmes in professional youth football: understanding the perceptions of programme deliverers. BMJ Open Sport Exerc Med. 2016;2(1):e000075.

Morgan EA, Johnson ST, Bovbjerg VE, Norcross MF. Associations between player age and club soccer coaches’ perceptions of injury risk and lower extremity injury prevention program use. Int J Sports Sci Coach. 2018;13(1):122–8.

Junge A, Lamprecht M, Stamm H, Hasler H, Bizzini M, Tschopp M, Reuter H, Wyss H, Chilvers C, Dvorak J. Countrywide campaign to prevent football injuries in Swiss amateur players. Am J Sports Med. 2011;39(1):57–63.

Lindblom H, Waldén M, Carlfjord S, Hägglund M. Implementation of a neuromuscular training programme in female adolescent football: 3-year follow-up study after a randomised controlled trial. Br J Sports Med. 2014;48(19):1425–1430.

Joy EA, Taylor JR, Novak MA, Chen M, Fink BP, Porucznik CA. Factors influencing the implementation of anterior cruciate ligament injury prevention strategies by girls football coaches. J Strength Condit Res. 2013;27(8):2263–2269.

Stoszkowski J, Collins D. Sources, topics and use of knowledge by coaches. J Sports Sci. 2016;34(9):794–802.

O'Brien J, Young W, Finch CF. The use and modification of injury prevention exercises by professional youth football teams. Scand J Med Science Sports. 2016. doi: 10.1111/sms.12756